# PeerJ

# Gooseneck barnacles (*Lepas* spp.) ingest microplastic debris in the North Pacific Subtropical Gyre

Miriam C. Goldstein[1,2] and Deborah S. Goodwin[3]

[1] Scripps Institution of Oceanography, University of California, San Diego, La Jolla, CA, USA
[2] California Sea Grant, La Jolla, CA, USA
[3] Sea Education Association, Woods Hole, MA, USA

## ABSTRACT

Substantial quantities of small plastic particles, termed "microplastic," have been found in many areas of the world ocean, and have accumulated in particularly high densities on the surface of the subtropical gyres. While plastic debris has been documented on the surface of the North Pacific Subtropical Gyre (NPSG) since the early 1970s, the ecological implications remain poorly understood. Organisms associated with floating objects, termed the "rafting assemblage," are an important component of the NPSG ecosystem. These objects are often dominated by abundant and fast-growing gooseneck barnacles (*Lepas* spp.), which predate on plankton and larval fishes at the sea surface. To assess the potential effects of microplastic on the rafting community, we examined the gastrointestinal tracts of 385 barnacles collected from the NPSG for evidence of plastic ingestion. We found that 33.5% of the barnacles had plastic particles present in their gastrointestinal tract, ranging from one plastic particle to a maximum of 30 particles. Particle ingestion was positively correlated to capitulum length, and no blockage of the stomach or intestines was observed. The majority of ingested plastic was polyethylene, with polypropylene and polystyrene also present. Our results suggest that barnacle ingestion of microplastic is relatively common, with unknown trophic impacts on the rafting community and the NPSG ecosystem.

## INTRODUCTION

Oceanic litter, termed "marine debris" or "plastic pollution," is a matter of increasing scientific and public concern (*STAP, 2011*; *US Environmental Protection Agency, 2011*; *Secretariat of the Convention on Biological Diversity and the Scientific and Technical Advisory Panel–GEF, 2012*). The durability and longevity that make plastic a useful substance also leads to its persistence in the marine environment, with consequences that include entanglement, damage to habitats, species transport, and ingestion (*National Research Council, 2008*). One study estimated that more than 267 species have been documented to ingest plastic (*Allsopp et al., 2006*), including mammals (*Eriksson & Burton, 2003*; *Jacobsen, Massey & Gulland, 2010*), seabirds (*Moser & Lee, 1992*; *Ryan, 2008*; *Van Franeker et al., 2011*), turtles (*Schuyler et al., 2013*), and a wide variety of fishes (*Possatto et al., 2011*;

Corresponding author
Miriam C. Goldstein,
miriam.goldstein@gmail.com

*Lusher, McHugh & Thompson, 2013*; *Anastasopoulou et al., 2013*). Negative effects of plastic ingestion may include intestinal blockage, diminished feeding stimulus, lowered steroid hormone levels, delayed ovulation and reproductive failure (*Azzarello & Van Vleet, 1987*; *Derraik, 2002*). Because oceanic plastic debris can contain high levels of hydrophobic toxins (*Endo et al., 2005*; *Frias, Sobral & Ferreira, 2010*; *Rios et al., 2010*; *Rochman et al., 2013*), ingestion of plastic debris may also increase toxic exposure (*Teuten et al., 2009*; *Gassel et al., 2013*).

Most plastic ingestion has been documented in vertebrates (Convention on Biological Diversity and STAP-GEF 2012), but the extent of plastic ingestion in marine invertebrates remains poorly known. Laboratory experiments suggest that many invertebrate species ingest plastic (reviewed in *Wright, Thompson & Galloway, 2013*). Suspended plastic particles (2–60 μm in diameter) were successfully fed to calanoid copepods, cladocerans, and salps in the context of studying particle size selectivity (*Burns, 1968*; *Wilson, 1973*; *Frost, 1977*; *Kremer & Madin, 1992*). In laboratory studies focused specifically on the incidence of plastic ingestion, plastic particles were readily consumed by an assortment of zooplankton (*Cole et al., 2013*) and benthic invertebrates (*Thompson et al., 2004*; *Browne et al., 2008*; *Graham & Thompson, 2009*; *Wegner et al., 2012*; *Von Moos, Burkhardt-Holm & Köhler, 2012*; *Besseling et al., 2013*). However, the evidence from natural ecosystems is far sparser. To date, we are aware of only three studies that have found *in situ* plastic ingestion in invertebrates: sandhopper amphipods (*Talitrus saltator*; *Ugolini et al., 2013*), Norway lobster (*Nephrops norvegicus*; *Murray & Cowie, 2011*), and flying squid (*Ommastrephes bartrami*; Day 1988 cited in *Laist, 1997*).

Though plastic pollution has been documented in the North Atlantic and North Pacific subtropical gyres since the early 1970s (*Carpenter & Smith, 1972*; *Wong, Green & Cretney, 1974*; *Day & Shaw, 1987*; *Moore et al., 2001*; *Law et al., 2010*; *Goldstein, Rosenberg & Cheng, 2012*), the ecological implications have been relatively little studied. In this open ocean ecosystem, the majority of marine debris are small particles (termed "microplastic," less than 5 mm in diameter) that float at the sea surface (*Hidalgo-Ruz et al., 2012*), though wind mixing moves some particles deeper (*Kukulka et al., 2012*). Floating plastics in these areas are primarily comprised of polyethylene, with polypropylene and polystyrene also present (*Rios, Moore & Jones, 2007*; *Goldstein, 2012*). Ingestion has been found in surface-feeding seabirds (*Fry, Fefer & Sileo, 1987*; *Avery-Gomm et al., 2012*) and epipelagic and mesopelagic fishes (*Boerger et al., 2010*; *Davison & Asch, 2011*; *Jantz et al., 2013*; *Choy & Drazen, 2013*), but the biota most likely to be impacted by microplastic pollution is the neuston, a specialized community associated with the air-sea interface which includes both zooplankton and substrate-associated rafting organisms (*Cheng, 1975*).

Rafting organisms in the open ocean are increasingly associated with floating plastic debris, which has supplemented natural substrates such as pumice and macroalgae (*Thiel & Gutow, 2005a*). Two species of lepadomorph barnacles (*Lepas anatifera* and *Lepas pacifica*) are widespread throughout the North Pacific Subtropical Gyre (NPSG) and frequently dominate the rafting assemblage (*Tsikhon-Lukanina, Reznichenko & Nikolaeva, 2001*). (A third species, *Lepas (Dosima) fascularis*, forms its own float at the end of the

juvenile stage and drifts independently, and is therefore not a major component of the rafting assemblage; *Newman & Abbott, 1980.*) These barnacles are omnivorous, feeding opportunistically on the neustonic zooplankton, and are said to "hold a singular position in having more sources of food to draw upon than any other organisms in the neuston (*Bieri, 1966*)." The barnacles are themselves preyed upon by omnivorous epipelagic crabs and the rafting nudibranch *Fiona pinnata* (*Bieri, 1966*; *Davenport, 1992*).

In this study, we hypothesized that *Lepas*' indiscriminate feeding strategy and position at the sea surface could cause this species to ingest microplastic, with unknown implications for NPSG ecology. To this end, we examined the gastrointestinal tracts of 385 *Lepas* from the NPSG for evidence of microplastic ingestion.

## METHODS

Floating debris items with attached gooseneck barnacles (Fig. 1A) were opportunistically collected during the 2009 Scripps Environmental Accumulation of Plastic Expedition (SEAPLEX) and two 2012 Sea Education Association (SEA) research cruises onboard the *SSV Robert C. Seamans*: S242, an undergraduate voyage from Honolulu, HI to San Francisco, CA (mid-June to mid-July 2012), and S243, the *Plastics at SEA: North Pacific Expedition* from San Diego, CA to Honolulu, HI (early October to mid-November 2012). Collection occurred by several means, including (1) from the vessel using a long-handled dip net (335 μm mesh, 0.5 m diameter mouth); (2) incidentally during neuston net (335 μm mesh, 0.5 × 1.0 m mouth) tows at the air-sea interface; and (3) from small boat surveys within 0.5 km of each vessel when sea conditions were calm. No specific permissions were required for these samples, since they were taken in international waters and did not involve protected species. Seven debris items were sampled on SEAPLEX and 29 by SEA (5 during S242 and 24 on S243). Stations within 8.5 km of each other were combined for a total of 19 sampling locations within in the northeastern Pacific Ocean (Fig. 2, Table 1).

During SEAPLEX, the entire piece of debris with attached barnacles was preserved in 5% Formalin buffered with sodium borate. When the item was too large to be preserved (e.g., a fishing buoy), barnacles were removed and preserved separately. On SEA cruises, as many barnacles as possible to a maximum of 50 were removed from the debris object and preserved in 10% ethanol. Where feasible, a fragment of the item itself was also sampled.

In the laboratory, capitulum length was measured using a ruler and species identification (*L. anatifera* or *L. pacifica*) determined for all intact individuals (Fig. 1B). Barnacles less than 0.8 cm were present, but not sampled in this study. Barnacles greater than approximately 0.8 cm in length were dissected and the contents of their stomach and intestinal tract examined under a dissecting microscope (6–25× magnification as needed). Barnacles were cut open with a scalpel, and the intestinal tract removed and placed in a separate section of the petri dish. The intestinal tract was opened lengthwise, and the contents examined systematically both visually and with forceps. To avoid cross-contamination between samples, each barnacle was dissected in a unique, clean petri dish and the scalpel was thoroughly rinsed with deionized water between each samples. Only microplastic

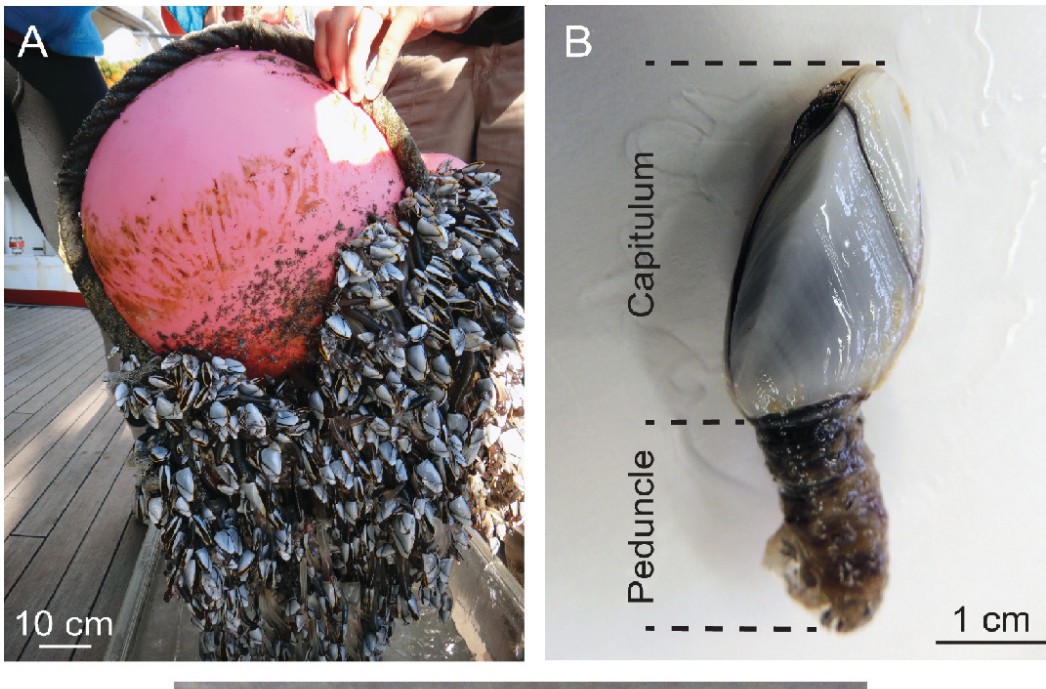

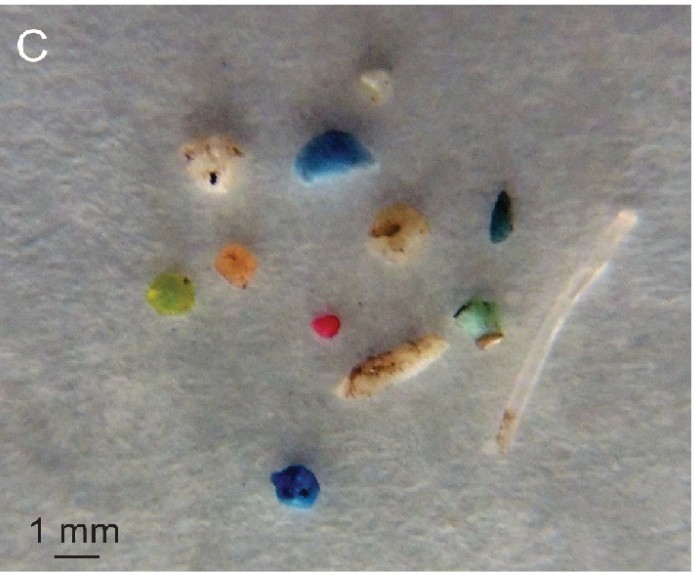

**Figure 1 Barnacles and ingestion microplastic.** (A) A dense aggregation of *Lepas* spp. barnacles growing on a buoy and attached line, collected in October 2012. (B) Basic anatomy of *Lepas* denoting the capitulum, which includes the body and its enclosing plates, and the peduncle, the muscular stalk that attaches the barnacle to the substrate. (C) Microplastic ingested by an individual barnacle.

fragments and monofilament that were clearly present inside the intestine were considered. Fine microfibers were discounted, as they could not be distinguished from airborne contamination. Because the vast majority of microplastic found were relatively large degraded fragments ($>0.5$ mm in diameter), visual examination was sufficient to confirm that the microplastic was present in the intestine, and not a result of contamination (Fig. 1C).

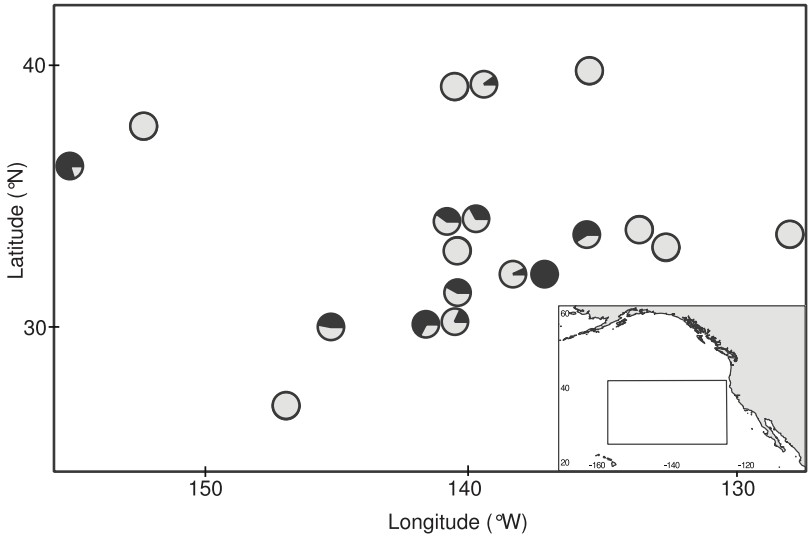

**Figure 2 Ingestion of microplastic by barnacles across the study area.** Circles indicate sampling stations and dark fill indicates the proportion of barnacles that had ingested microplastic at each site. Station coordinates, sample sizes, and ingestion proportions are given in Table 1.

**Table 1 Station locations and proportion of microplastic ingestion.**

| Station ID | Date of collection | Latitude (°N) | Longitude (°W) | Total no. barnacles | Proportion with plastic | Proportion without plastic |
|---|---|---|---|---|---|---|
| S242-021-DN | 1-Jul-12 | 36.135 | 154.957 | 5 | 0.80 | 0.20 |
| S242-023-DN | 2-Jul-12 | 37.672 | 152.163 | 20 | 0.00 | 1.00 |
| S243-083-DN | 31-Oct-12 | 27.000 | 146.782 | 10 | 0.00 | 1.00 |
| S243-069-DN | 27-Oct-12 | 30.057 | 145.057 | 15 | 0.47 | 0.53 |
| S243-055-057-058-DN | 24-Oct-12 | 30.140 | 141.220 | 80 | 0.68 | 0.33 |
| S243-051-052-DN | 23-Oct-12 | 30.230 | 140.690 | 34 | 0.18 | 0.82 |
| U39.F32 | 15-Aug-09 | 34.076 | 140.474 | 53 | 0.40 | 0.60 |
| S243-046-DN | 22-Oct-12 | 31.330 | 140.338 | 52 | 0.42 | 0.58 |
| S3.F6 | 10-Aug-09 | 32.911 | 140.320 | 2 | 0.00 | 1.00 |
| S242-031-NT | 6-Jul-12 | 39.178 | 140.160 | 12 | 0.00 | 1.00 |
| S4.F30-F26 | 14-Aug-09 | 34.090 | 139.870 | 9 | 0.33 | 0.67 |
| S242-032-DN | 6-Jul-12 | 39.270 | 139.570 | 10 | 0.10 | 0.90 |
| S2.F22-U40.F11 | 9-Aug-09 | 32.050 | 137.928 | 15 | 0.07 | 0.93 |
| F13 | 9-Aug-09 | 32.075 | 137.223 | 1 | 1.00 | 0.00 |
| S243-032-DN | 16-Oct-12 | 33.563 | 135.432 | 17 | 0.59 | 0.41 |
| S242-035-DN | 8-Jul-12 | 39.717 | 135.325 | 10 | 0.00 | 1.00 |
| S243-025-027-DN | 14-Oct-12 | 33.700 | 133.460 | 13 | 0.00 | 1.00 |
| S243-023-DN | 13-Oct-12 | 33.051 | 132.445 | 14 | 0.00 | 1.00 |
| S243-011-DN | 9-Oct-12 | 33.493 | 127.715 | 13 | 0.00 | 1.00 |

Plastic particles found in the stomach or intestine were quantified, photographed digitally against a ruler for size assessment, rinsed with fresh water and stored in a glass vial for later analyses. The maximum diameter (feret diameter) and two-dimensional area of each particle were digitally measured with the software package NIH ImageJ (*Schneider, Rasband & Eliceiri, 2012*). On the SEAPLEX cruise in 2009, we also measured the diameter and area of all plastic particles captured in surface-towed plankton nets (number of particles = 30,518) using NIH ImageJ-based tools in the Zooprocess software calibrated against manual measurements (*Gilfillan et al., 2009*; *Gorsky et al., 2010*).

We identified the type of plastic recovered from a randomly selected subset of barnacles (Barnacles $N = 42$; particles $N = 219$). A Raman spectrometer (PeakSeeker Pro-785 with AmScope operated at 10–50 mW and 5–20 s integration time; Raman Systems MII, Inc/Agiltron, Inc., Woburn, MA) and associated RSIQ software were used to identify plastic type. The Raman spectrum for each plastic piece was compared to a reference library of known plastic types for identification. Particles of clear, white, gray and pale-colored (light blues and greens, oranges and yellows) plastics yielded high quality Raman spectra and were readily identifiable. Those that were darker (medium to dark blues, reds and greens as well as black; 35% of particles subjected to Raman spectroscopy) were heated by the laser beam and melted even at the lowest possible power and integration time settings, resulting in no usable spectra. We also identified a subset of the debris items to which the barnacles were attached. Fragments of 18 objects were collected for analysis, but 6 could not be identified due to darker pigmentation which caused melting under the laser.

Statistics and figures were generated with the R statistical environment, version R-2.15.1 (*R Development Core Team, 2012*) and QuantumGIS, version 1.8.0-Lisboa (*QGIS Development Team, 2013*).

## RESULTS

Of the 385 barnacles examined, 129 individuals (33.5%) had ingested plastic (Fig. 2, Table 1). These included 243 *Lepas anatifera* and 85 *Lepas pacifica* (57 barnacles could not be identified to species), of which 90 *L. anatifera*, 34 *L. pacifica*, and 5 *Lepas* spp. contained plastic. Forty-one of the barnacles that ingested plastic had one plastic particle in their stomach or intestines, 26 individuals had two particles, and 57 individuals contained three or more particles, to a maximum of 30 particles (Fig. 3A).

Overall, the number of ingested particles was positively correlated to capitulum length (Kendall's tau = 0.099, $p = 0.015$). However, when we considered only barnacles that had ingested plastic, the correlation between plastic ingestion and capitulum length was not significant (Kendall's tau = $-0.080$, $p = 0.229$). Individuals with a capitulum length between 2 and 3 cm consumed the greatest number plastic particles (Fig. 3B). With the exception of one individual, all the barnacles that consumed plastic had a capitulum length of 1.7 cm or greater.

In total, 518 plastic particles were recovered from barnacle digestive tracts. Of these, 99% were degraded fragments and 1% were monofilament line. None of the pre-production pellets known as "nurdles" were found. The median diameter of ingested

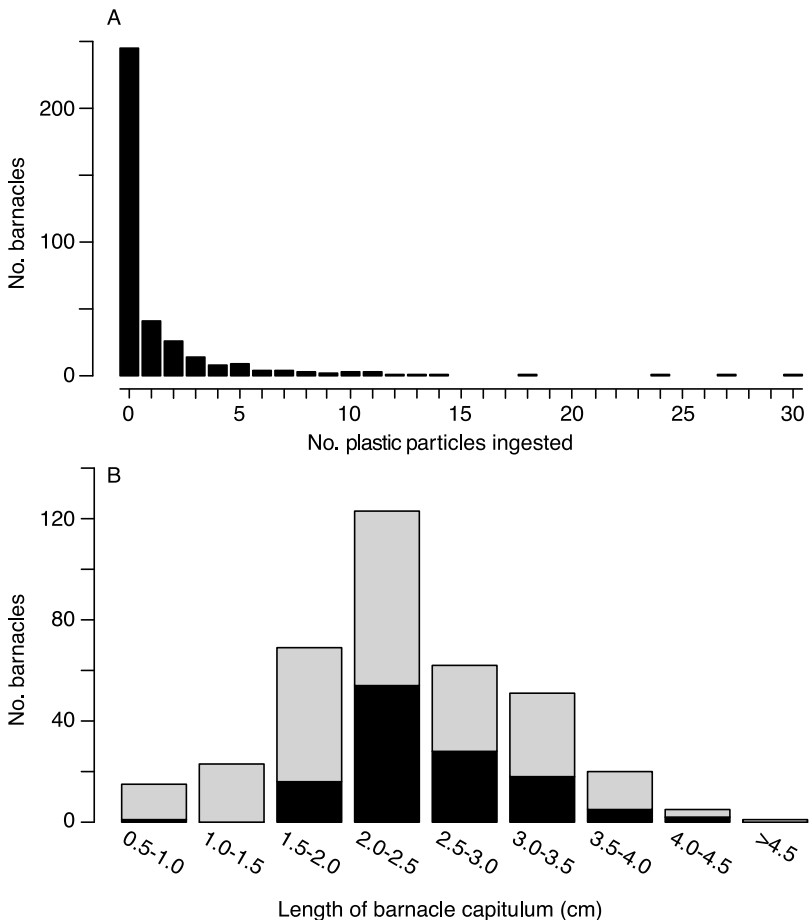

**Figure 3** **Number of microplastic particles ingested by barnacles.** (A) Frequency distribution of microplastic pellets ingested by individual lepadid barnacles ($N = 385$). (B) Frequency distribution of ingestion by capitulum length ($N = 369$; sample size is smaller than above since capitulum length was not measured for 16 barnacles). Black bars are the number of individual barnacles that ingested plastic and grey bars are the number of individual barnacles that did not ingest plastic. Bins of capitulum length are greater than the first value, and less than or equal to the second value (e.g., $>0.5$ cm and $<= 1.0$ cm). Percentages of ingestion by size class are as follows: 6.7%, 0, 23.2%, 43.9%, 45.2%, 35.3%, 25.0%, 40.0%, 0.

particles was 1.41 mm, and the median surface area 1.00 mm$^2$, smaller than the median diameter of 1.78 mm and median surface area of 1.27 mm$^2$ for all particles collected in nets during 2009 (Fig. 4, Kolmogorov–Smirnov test $p < 0.001$). The smallest particle ingested by barnacles had a maximum diameter of 0.609 mm and the largest (a long thin fragment) a maximum diameter of 6.770 mm. No blockage of the stomach or intestine was observed, and particles did not accumulate in any area of the digestive tract. All particles were of a plausible size to pass through the anus.

Of the randomly selected subset of 219 ingested plastic particles that were analyzed for plastic type, 58.4% were polyethylene, 5.0% were polypropylene, and 1.4% were polystyrene. As noted in the Methods section, we were unable to identify 35% of the

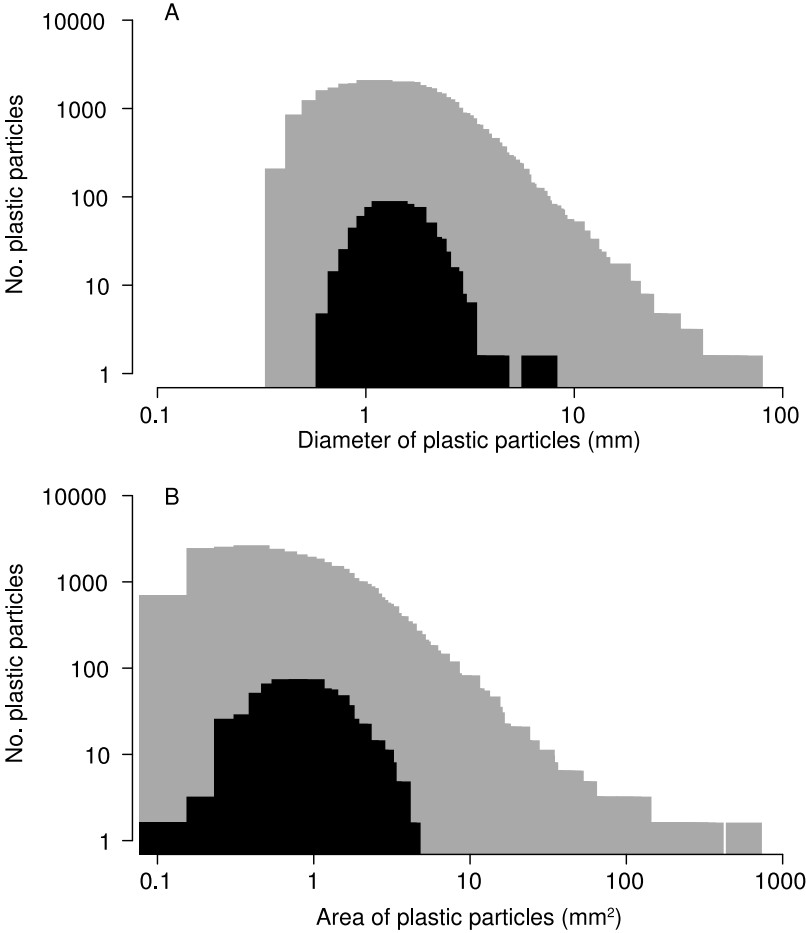

**Figure 4 Size of microplastic particles ingested by barnacles.** Size–frequency distributions for (A) maximum diameter and (B) two-dimensional surface area of particles ingested by barnacles (black; $N = 507$) compared to of all microplastic particles collected in 2009 (grey; $N = 30{,}518$). Note: 518 microplastic particles were recovered from barnacles, but 11 were lost before they could be photographed for this analysis.

subset due to darker pigmentation in these particles, which caused melting under the Raman spectrophotometer. Of the 29 barnacles that had ingested more than one piece of plastic, 66% contained more than one type of plastic. The plastic types of 12 floating debris items to which barnacles were attached were more diverse than those of ingested particles. Four substrates were polystyrene, 3 were polyethylene, 2 were polypropylene, 2 were polyethylene terephthalate, and one was tire rubber.

## DISCUSSION

Our results show that 33.5% of lepadid barnacles collected from the NPSG ingested microplastic, and that the sizes and types of ingested particles were approximately representative of microplastic found on the NPSG surface. Plastic ingestion in these barnacles may therefore be explained by non-selective suspension feeding while exposed to high concentrations of microplastic.

The percentage of barnacles observed with ingested plastic in this study is higher than the 9.2% found in NPSG micronektonic fishes (*Davison & Asch, 2011*) and the 19–24.5% found in larger mesopelagic fishes (*Jantz et al., 2013*; *Choy & Drazen, 2013*). It is likely that barnacles encounter microplastic more frequently than vertically migrating mesopelagic fishes due to the barnacles' consistent location at the air-sea interface. Since barnacles probably clear their guts in a matter of hours (*Ritz, 2008*), it is likely that a higher percentage of the barnacle population interacts with microplastic than is presented here. Unfortunately, due to logistical considerations on both cruises, barnacle samples were not usually concurrent with neuston tows. Since neustonic microplastic is highly spatially heterogeneous (*Ryan et al., 2009*), we are thus unable directly compare neustonic microplastic concentrations with incidence of barnacle ingestion. However, the sampling area is known to contain high concentrations of neustonic microplastic (*Moore et al., 2001*; *Goldstein, Rosenberg & Cheng, 2012*).

The objects to which the barnacles are attached may also shed microplastic particles, increasing the likelihood of those particles being ingested by the local rafting community. However, the microplastics ingested by individual barnacles in this study were of multiple plastic types and colors, suggesting they are taking in particles from the surrounding water rather than solely from the substrate to which they are attached. Lepadid barnacles are known to be very nonselective feeders. For example, *L.anatifera* opportunistically ingests a wide variety of zooplankton and even fills its gut with sand when stranded on the beach (*Howard & Scott, 1959*). *L. anatifera* can also readily consume large prey items up to 5 mm in diameter, larger than the majority of microplastic debris (*Patel, 1959*). Less is known about the feeding habits of *L. pacifica*, but it is presumed to have a similar feeding ecology as *L. anatifera* and other lepadid barnacles (*Crisp & Southward, 1961*; *Cheng & Lewin, 1976*). To avoid difficulties in identifying plastic with darker pigmentation, future studies might consider supplementing Raman spectroscopy with density analysis (*Moret-Ferguson et al., 2010*), or utilizing Fourier transform infrared spectroscopy when available (*Rios, Moore & Jones, 2007*; *Goldstein, 2012*).

We found only one barnacle with a capitulum length of less than 1.7 cm that had ingested plastic. This observation implies that barnacles may need to reach a certain size before plastic ingestion is possible, perhaps due to the size of the cirri or oral opening. However, our study used visual methods to identify microplastic in barnacle gut contents, and spectroscopic methods or chemical digestion of the tissue are needed to positively identify plastic particles smaller than approximately 300 μm (*Claessens et al., 2011*; *Hidalgo-Ruz et al., 2012*). It is therefore possible that plastic ingestion in the smaller barnacles was not detected in this study.

Assessing the ecological significance of plastic ingestion in pelagic invertebrates and fishes remains a challenge. Even in relatively well-studied species, it has been difficult to link plastic ingestion to mortality. For example, two studies of Laysan and black-footed albatross chicks did not find a linkage between cause of death and plastic ingestion (*Sileo, Sievert & Samuel, 1990*; *Sievert & Sileo, 1993*), though a third study linked plastic ingestion with lower body weight in adult birds (*Spear, Ainley & Ribic, 1995*). Most

studies on invertebrates have been relatively short-term investigations that have not found acute negative effects (*Thompson et al., 2004*; *Browne et al., 2008*; *Graham & Thompson, 2009*), with the exception of an inflammatory immune response in mussels (*Von Moos, Burkhardt-Holm & Köhler, 2012*). In zooplankton, the presence of non-edible particles can reduce the rate of feeding on edible particles (*Huntley, Barthel & Star, 1983*; *Ayukai, 1987*; *Cole et al., 2013*), and physical interference with sensory apparatus may occur in very high-plastic environments (*Cole et al., 2013*). The lepadid barnacles in this study did not show evidence of acute harm (e.g., intestinal blockage or ulceration), though negative long-term effects cannot be ruled out.

Plastic ingestion may also lead to increased body loads of persistent organic pollutants in both vertebrates and invertebrates (*Teuten et al., 2009*; *Yamashita et al., 2011*; *Gassel et al., 2013*; *Besseling et al., 2013*), but it is not known whether this occurs in barnacles, or has population-level ramifications in any taxa (*Gouin et al., 2011*). For example, a modeling study based on lugworms (*Arenicola marina*) did not find a significant toxicological risk from plastic-adsorbed pollutants (*Koelmans et al., 2013*). Because *L.anatifera* appear to survive well in the laboratory (*Patel, 1959*), more detailed studies may be possible.

If barnacles are an important prey item, it is possible that their ingestion of plastic particles could transfer plastic or pollutants through the food web. Plastic particles found in fur seals (*Eriksson & Burton, 2003*), piscivorous fishes (*Davison & Asch, 2011*), and crabs (*Farrell & Nelson, 2013*) have been linked to consumption of contaminated prey. The only documented predator of rafting *Lepas* spp. is the nudibranch *Fiona pinnata* (*Bieri, 1966*), though it is probable that omnivorous rafting crabs also consume barnacles to some extent (*Davenport, 1992*; *Frick et al., 2011*). Relatively low rates of predation on these barnacles may explain *Lepas'* place as one of the most abundant members of the North Pacific subtropical rafting community (*Newman & Abbott, 1980*; *Thiel & Gutow, 2005b*). For example, one study found that *L. pacifica* was excluded from nearshore kelp forests by the fish *Oxyjulis californica*, but was able to inhabit floating kelp paddies in high densities when *O. californica* was absent (*Bernstein & Jung, 1979*). Studies of the diets of fishes associated with Fish Aggregating Devices (FADs) have found that fishes associated with floating objects rarely feed directly on the fouling community (*Ibrahim et al., 1996*; *Nelson, 2003*; *Vassilopoulou et al., 2004*). The likelihood of predators ingesting plastic by feeding on barnacles may therefore be relatively low.

While plastic ingestion in taxa such as sea turtles (*Schuyler et al., 2013*) and cetaceans (*Jacobsen, Massey & Gulland, 2010*) is clearly detrimental, the implications of widespread plastic ingestion in *Lepas* remain uncertain. Since little is known about the trophic structure and connectivity of both the rafting and drifting components of the neuston, additional studies are necessary to determine the impacts of microplastic ingestion on the rafting community and the larger pelagic ecosystem.

## ACKNOWLEDGEMENTS

We thank the captains, crews, and students of the SEAPLEX cruise on the R/V *New Horizon* and Sea Education Association cruises S-242 and S-243 on the SSV *Robert C. Seamans*. Assistance from L Sala, MD Ohman, and E Zettler made this project possible. The manuscript was improved by helpful comments from reviewers M Cole and D Rittschof.

### Funding

Funding for the SEAPLEX cruise was provided by University of California Ship Funds, Project Kaisei/Ocean Voyages Institute, AWIS-San Diego, and NSF IGERT Grant No. 0333444. MCG was supported by the California Department of Boating and Waterways Contract 05-106-115, NSF GK-12 Grant No. 0841407 and donations from Jim & Kris McMillan, Jeffrey & Marcy Krinsk, Lyn & Norman Lear, Ellis Wyer, and Petersen Charitable Foundation. Laboratory supplies, support, and some analytical equipment were supplied by the SIO Pelagic Invertebrate Collection, the California Current Ecosystem LTER site supported by NSF, and Sea Education Association. The funders had no role in study design, data collection and analysis, decision to publish, or preparation of the manuscript.

### Grant Disclosures

The following grant information was disclosed by the authors:
NSF Grant: Nos. 0333444, 0841407.
California Department of Boating and Waterways Contract 05-106-115.

### Competing Interests

Deborah S. Goodwin is an employee of the Sea Education Association.

### Author Contributions

- Miriam C. Goldstein and Deborah S. Goodwin conceived and designed the experiments, performed the experiments, analyzed the data, contributed reagents/materials/analysis tools, wrote the paper.

### Data Deposition

The following information was supplied regarding the deposition of related data:
CCE LTER Datazoo http://oceaninformatics.ucsd.edu/datazoo/data/ccelter/datasets.

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
