# Peer review of "Gooseneck barnacles (Lepas spp.) ingest microplastic debris in the North Pacific Subtropical Gyre"

_PeerJ, doi:10.7717/peerj.184_

## Round 0.1 · original submission · Minor Revisions

The issues raised by both reviewers need to be carefully addressed before this ms can be accepted for publication. Please revise your ms carefully and submit your revised version, together with a detailed response to reviewers' comments in a format of point-to-point.

·

Basic reporting

Overall

This paper considers whether microplastics are ingested by barnacles that reside of floating debris in the North Pacific gyre. There are few studies of in-situ organisms ingesting microplastics so the data presented is a particularly important addition to the literature. The introduction is clear and well-informed, the methods and results clear and concise, and the discussion comprehensive. Overall I consider this a well-informed and highly interesting paper that I would be happy to recommend for publication.

Experimental design

Methodology

Line 85-86 – Dissection procedure could do with a little more detail (e.g. how was intestine removed? Was it cut open longitudinally? Was observation for microplastic debris systematic? Was the intestine washed through or plastics picked off? What was the magnification used?). Further, were any protocols included to mitigate contamination of samples (i.e. airborne or introduced via scalpel or clothing for example)? If not, perhaps a sentence to justify why this was not required in this study?

Line 92 – Does (N=30,518) refer to number of tows (as positioning of text suggests), or number of plastic particles collected?

Validity of the findings

Results

Figure 1 – did you consider comparing the sites used with number of microplastic particles found in the seawater trawls?

Line 117-119 – Looking at Figure 2b this correlation is not particularly evident; instead of looking at a correlation, I could see a histogram working much better here (similar to used in Fig2a). If you have data from the seawater trawls, would you be able to compare number of particles ingested Vs number of particles found in trawls around location the barnacle was sampled from?

Line125-127 – What types of plastic dominated the seawater trawls? (i.e. were there lots of fibres or nurdles present in the seawater that the barnacles were not eating?)



Discussion

In method (Line 103) Raman laser melted some plastics. There is no follow-up on alternate method that might have been used instead?

Additional comments

General Comments

While most readers will be familiar with what a barnacle looks like, “capitulum” is a very specific term that will not be well known to the majority of readers. Perhaps a small figure with an image of one of your barnacle specimens with some basic annotations would be appropriate? Would also be interesting to see a photo of some of the plastics recovered.

·

Basic reporting

fine

Experimental design

well explained

Validity of the findings

valid for over half the particles.

Additional comments

This is a well done offering. There are several things to consider with this paper:
How long do particles stay in a barnacle gut? Are you really measuring just a snapshot of the percentabge of barnacles that ate particles in the last x hours? Is ingested the best word to use, or would a phrase like found in their guts be more appropriate. It is clear that fish eat Lepas, it is very likely that there is considerable predation on Lepas in the open ocean. Do you believe for example that oceanic Lepas die of old age? Ms could be improved by addressing these issues and at least commenting on alternative methods that might be used to determine what kinds of plastic the colored particles that melt are made out of. Have the macroplastics been characterized? How might they compare.

---

## Round 0.2 · accepted · Accept

I am very happy to know that you have addressed most issues raised by the reviewers and I am happy to accept this ms for publication. Thank you for choosing PeerJ as outlet of your research findings.